# PM10 and Other Climatic Variables Are Important Predictors of Seasonal Variability of Coccidioidomycosis in Arizona

Daniel R. Kollath,[a] Joseph R. Mihaljevic,[b] (iD) Bridget M. Barker[a]

[a]Pathogen and Microbiome Institute, Northern Arizona University, Flagstaff, Arizona, USA
[b]School of Informatics, Computing, and Cyber Systems, Northern Arizona University, Flagstaff, Arizona, USA

**ABSTRACT** Coccidioidomycosis (Valley fever) is a disease caused by the fungal pathogens *Coccidioides immitis* and *Coccidioides posadasii* that are endemic to the southwestern United States and parts of Mexico and South America. Throughout the range where the pathogens are endemic, there are seasonal patterns of infection rates that are associated with certain climatic variables. Previous studies that looked at annual and monthly relationships of coccidioidomycosis and climate suggest that infection numbers are linked with precipitation and temperature fluctuations; however, these analytic methods may miss important nonlinear, nonmonotonic seasonal relationships between the response (Valley fever cases) and explanatory variables (climate) influencing disease outbreaks. To improve our current knowledge and to retest relationships, we used case data from three counties of high endemicity in southern Arizona paired with climate data to construct a generalized additive statistical model that explores which meteorological parameters are most useful in predicting Valley fever incidence throughout the year. We then use our model to forecast the pattern of Valley fever cases by month. Our model shows that maximum monthly temperature, average PM10, and total precipitation 1 month prior to reported cases (lagged model) were all significant in predicting Valley fever cases. Our model fits Valley fever case data in the region of endemicity of southern Arizona and captures the seasonal relationships that predict when the public is at higher risk of being infected. This study builds on and retests relationships described by previous studies regarding climate variables that are important for predicting risk of infection and understanding this fungal pathogen.

**IMPORTANCE** The inhalation of environmental infectious propagules from the fungal pathogens *Coccidioides immitis* and *Coccidioides posadasii* by susceptible mammals can result in coccidioidomycosis (Valley fever). Arizona is known to be a region where the pathogen is hyperendemic, and reported cases are increasing throughout the western United States. *Coccidioides* spp. are naturally occurring fungi in arid soils. Little is known about ecological factors that influence the growth of these fungi, and a higher environmental burden may result in increases in human exposure and therefore case rates. By examining case and climate data from Arizona and using generalized additive statistical models, we were able to examine the relationship between disease outbreaks and climatic variables and predict seasonal time points of increased infection risk.

**KEYWORDS** Valley fever, human fungal pathogen, GAM, climate, coccidioidomycosis, *Coccidioides*, disease ecology, generalized additive model, mycology

Over the past 40 years, the incidence of invasive fungal infections has dramatically increased, and the number of at-risk populations, such as those who are immunocompromised, has risen concurrently (1). Medically relevant fungal infections are emerging across the globe, with a wide range of disease outcomes from asymptomatic infections to systemic mycoses requiring lifelong treatment. One medically relevant

Address correspondence to Bridget M. Barker, Bridget.Barker@nau.edu.

The authors declare no conflict of interest.

group of organisms are the dimorphic fungi in the order Onygenales (2). This group includes human and animal pathogens such as *Emergomyces*, *Emmonsiellopsis*, *Emmonsia*, *Blastomyces*, and *Histoplasma* species complexes and the two species in the genus *Coccidioides*. The organisms in this order are known to cause subclinical to clinical diseases in humans as well as other mammals (2). Most fungal infections are environmentally acquired, yet few studies have assessed the ecological factors that may influence fungal transmission dynamics.

Coccidioidomycosis, commonly called Valley fever, is a fungal respiratory disease caused by *Coccidioides immitis* and *Coccidioides posadasii* that is known to infect mammals throughout the southwestern United States and semiarid to arid areas of Mexico, Central America, and South America (3, 4). *Coccidioides immitis* in found primarily in California and Washington state, and *Coccidioides posadasii* is found in all other regions of endemicity. These fungi have two distinct life cycles, an environmental life stage growing as vegetative mycelia in soil and a parasitic stage where the fungus undergoes a dramatic morphological shift into a spherule within the host (5). Infection occurs when a susceptible host inhales airborne spores. The pulmonary infection may result in symptoms such as fever, cough, night sweats, chest distress, and fatigue and can disseminate from the lungs to cause more severe disease outcomes such as meningitis (6). In areas of endemicity, Valley fever presents a significant public health risk. From 2011 to 2017 there were approximately 95,000 cases of Valley fever reported to the Centers for Disease Control and Prevention (CDC) in 26 different states with an average of 200 deaths every year, although due to inconsistent reporting, it is estimated that the actual number of cases is 6 to 14 times greater (7, 8).

The biotic and abiotic ecological factors that influence the distribution and environmental burden of *Coccidioides* spp. are not well understood due to difficulties in detecting and isolating the fungus from soil (9–12). Previous studies have shown weak correlations between soil properties such as alkaline pH and salinity and the presence of the fungus (9, 10, 13, 14). Certain meteorological parameters, such as precipitation and temperature, likely influence the fluctuation of fungal concentrations in the soil and air, which would lead to increases in infections at certain times of the year. Comrie showed evidence of bimodal seasonality of Valley fever cases that coincided with the precipitation regime of the Sonoran Desert and has assessed the role of dust storms (haboobs)—which appear to play no direct role in the incidence of Valley fever (15, 16). Talamantes et al. found no to weak statistical evidence of climatic factors driving Valley fever cases and attributed fluctuations to human activity (17). A case report in Arizona showed that linear statistics may not be the best approach to link climatic variables as well as PM10 to fluctuations in Valley fever cases (18). These previous approaches to examine the impact of climate on Valley fever case variabilities have been informative; however, with the growing number of cases we must examine if these patterns hold and develop methods that can define climate variables that predict seasonal or yearly outbreaks of Valley fever.

Accurately predicting an increase in Valley fever cases, and determining which months are likely to have high fungal burdens in the environment, will help health officials promote public safety in regions of endemicity. Here, we build statistical models that explain how climate variables are associated with seasonally recurrent human disease. Specifically, we fit generalized additive models (GAMs) to reported human case data to examine infection patterns of Valley fever in the areas of hyperendemicity with the greatest numbers of reported cases in the United States and a known prevalence of *Coccidioides posadasii* in the soil, namely, Pima, Pinal, and Maricopa counties in Arizona, using climatic variables as covariates.

Our goal is to understand the seasonal dynamics of human infections and how interannual variation in climate will affect the risk of increased cases by retesting what others have previously observed with new data and analytical methods. We chose our climatic covariates based on previous studies, as well as our own hypotheses.

**TABLE 1** Predictive performance statistics of different models used to predict seasonal Valley fever cases[a]

| Model name | RMSE | R-sq. (adj) | Deviance explained (%) | AIC | Δ AIC |
|---|---|---|---|---|---|
| A. No seasonality | 144.32 | 0.30 | 31.10 | 2,683.43 | 740.7 |
| B. Precipitation | 63.72 | 0.85 | 93.70 | 2,073.62 | 130.89 |
| C. Precipitation/temp | 63.81 | 0.85 | 93.80 | 2,072.88 | 130.15 |
| D. All variables | 47.90 | 0.92 | 95.40 | 1,985.71 | 42.98 |
| E. Lagged[b] | 42.72 | 0.93 | 96.20 | 1,942.73 | 0.00 |

[a]Model performance is evaluated on four different metrics. The best-fit model should have the lowest error (root mean squared error [RMSE]), best fit {measured in adjusted $R$-squared [R-sq. (adj)]}, highest explained deviance, and lowest AIC. Random effects of county and year are included in all models.
[b]This is the model with the best fit.

Temperature and precipitation are likely biologically important to the life cycle of the fungus (15, 19–21).

Inclement weather, such as dust storms (haboobs), has been proposed to increase the risk of infection by *Coccidioides* (22, 23). Although recent analysis shows no such connection (16), we still expect wind speed may be important for explaining human disease because of the dispersal of spores that establish infection (16, 17, 24, 25). Moreover, we hypothesize that the concentration of inhalable particulate matter, quantified as PM10, is important because infectious spores could be dispersed via dust. Finally, we accounted for delays in diagnosis by introducing lagged effects because there is a delay between infection and symptom onset. These covariates were put into models and evaluated for importance for model performance; the highest-performing model was used for the final analysis.

We hypothesize that (i) seasonal temperature will impact fungal growth and development, which may be observed by increased cases with higher temperature in the summer months, and (ii) precipitation will impact soil moisture and fungal growth, and thus, greater total precipitation in the winter and during the monsoon season may lead to increased infections, but this effect could be delayed until the soil dries out and conidia can be more efficiently dispersed by wind.

## RESULTS

Human cases of Valley fever in Maricopa, Pima, and Pinal counties, Arizona, from 2013 to 2018 demonstrate seasonality. There is a pattern of bimodal seasonality in the three sampled Arizona counties (see Table S1 in the supplemental material). In other words, Valley fever cases fluctuate throughout the year based on climate. Specifically, cases tend to rise during the summer months and winter months and decrease during the spring but show a slight increase in the fall months.

The Lagged Model that included monthly precipitation, mean maximum monthly temperature, average monthly wind speed, mean monthly PM10, lagged precipitation, and lagged PM10 was the most supported model that best explained seasonal patterns of Valley fever (Tables 1 and 2). The effectiveness of the model to predict the future was assessed using 2019 Valley fever case data (Fig S2). This model has high within-sample predictive ability, with an $R^2$ of 0.93 and explained deviance of 96.2%. Specifically, the model suggests that there are significant seasonal effects of PM10, mean maximum temperature, lagged precipitation, and lagged PM10.

**Average monthly PM10.** Of the climate variables used in our study, the inhalable particulate matter 10 $\mu$m and below (PM10) shows the strongest effects on Valley fever cases. Specifically, high PM10 in the winter (November to January) is associated with higher detection of human cases in the winter (Fig. 1). The effect is particularly strong in December and January. However, during the rest of the year the effect of PM10 is minimal (Fig. 1). Figure 1A explores the relationship between month and PM10 and how the combination affects case numbers. The model is predictive across a range of

**TABLE 2** Covariate effects of the lagged model[a]

| Covariate | EDF | P value |
|---|---|---|
| Max temp | 2.215 | 0.0214 |
| PM10 | 7.24 | <0.0001 |
| Lagged precipitation | 2.80 | 0.0003 |
| Lagged PM10 | 3.05 | 0.0095 |
| Wind speed | 0.00013 | 0.67 |
| Precipitation | 1.0003 | 0.89 |

[a]Effective degrees of freedom (EDF) of each covariate represents the complexity of each smooth term; the greater values are more complex (1.00 = linear). P value represents overall significance of the smooth term.

PM10 values. The contour lines are identifying combinations of PM10 values and month and the relationship to Valley fever cases (26).

**Mean monthly maximum temperature.** In January and February lower maximum temperatures are associated with increased infections, although the effect is weaker than the effect of PM10. Starting in the early summer months, there is a stronger effect, such that lower temperatures are associated with increased cases (Fig. 2). In other words, during the summer to late fall, when temperatures are cooler than average, we tend to see more Valley fever cases.

**Lagged precipitation and PM10.** Precipitation and PM10 each have significant lagged effects. In other words, the values of these variables 2 months prior to disease reporting have effects on the number of Valley fever cases detected in the current month. First, we tend to see more cases in the late fall and winter months when there was lower precipitation 2 months before (Fig. 3), likely indicating lower overall moisture for the preceding 2 months. Second, lagged PM10 showed more complex effects, although these effects are generally weaker than current PM10 (above). In the winter and summer months, low lagged PM10 has a weakly positive effect, associated with higher cases; however, this effect is reversed in the spring months, such that high PM10 in the late winter is associated with higher cases in the spring months (Fig. 4).

## DISCUSSION

**Seasonal effects on Valley fever incidence in Arizona.** This study found significant monthly effects of climate on cases of coccidioidomycosis. PM10 shows the strongest effect on the number of cases when PM10 was higher during the winter months leading to increased infections. Interestingly, PM10 during the rest of the year does not appear to have a strong effect on infections. Other variables were also significant in predicting Valley fever cases, although not as strong as PM10. Decreased mean maximum temperatures in January and February are associated with increased

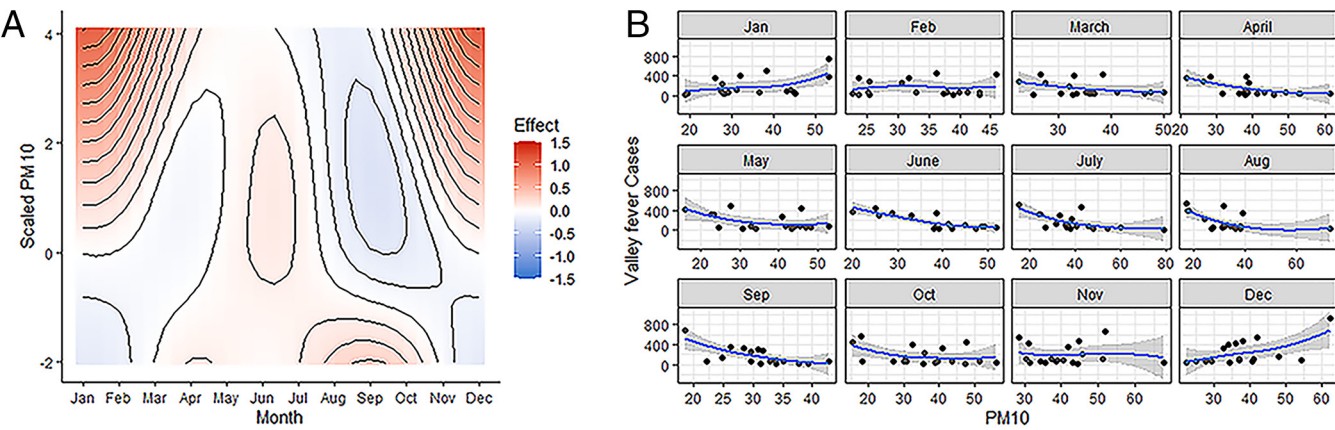

**FIG 1** The effects of PM10 on the number of Valley fever cases. (A) Contour plot of predicted Valley fever cases across the entire range of PM10 values (micrograms per cubic meter). Red means that there is a strong positive effect of PM10 on cases (i.e., increased infections), and blue means that there is a weak to negative effect of PM10 on cases (i.e., reduced infections). (B) Raw monthly PM10 data plotted against monthly Valley fever data added together from all three counties. Locally estimated scatterplot smoothing (LOESS) smoother line used in order to visualize relationships.

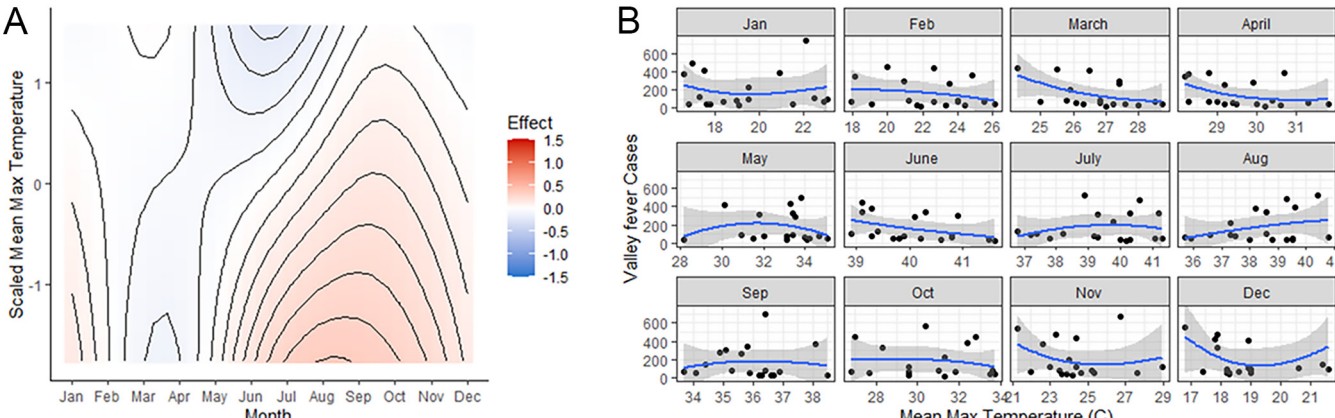

**FIG 2** The effects of temperature on the number of Valley fever cases. A 2-month lag time before the month where cases were reported was employed. (A) Simulation of predicted Valley fever cases across the entire range of mean maximum temperatures. Red means that there is a strong effect of maximum temperatures and Valley fever infections, and blue indicates a weak effect of maximum temperature on Valley fever infections. (B) Raw mean maximum monthly temperature data plotted against monthly Valley fever data added together from all three counties. LOESS smoother line used in order to visualize relationships.

infections; however, the effect becomes weaker in early summer when there is an even stronger effect of cooler maximum temperatures associated with increased cases. These contemporary results coincide with previous studies that examined the effect of PM10 on Valley fever cases. Comrie in 2005 as well as others showed that PM10 is positively associated with Valley fever rates in the winter months but is not a useful predictor during monsoon season in Arizona (15, 27). This study shows results contradictory to previous studies done in California where negative relationships were found between dust and Valley fever and no significance in the winter months (23, 28). These differences highlight the dissimilarities between the two species of *Coccidioides* and the subsequent disease dynamics that follow infection.

We also detected significant lagged effects on both PM10 and precipitation. First, there appears to be a trend of increased infections in the fall and winter months when precipitation was low 2 months prior. Second, in the winter and summer months, low lagged PM10 has a weakly positive effect, associated with higher cases; however, this effect is reversed in the spring, such that high PM10 in the late winter months is associated with higher cases in the spring months. Thus, temperature and precipitation do not have as much of an effect on cases as we hypothesized, but rather, PM10 is the

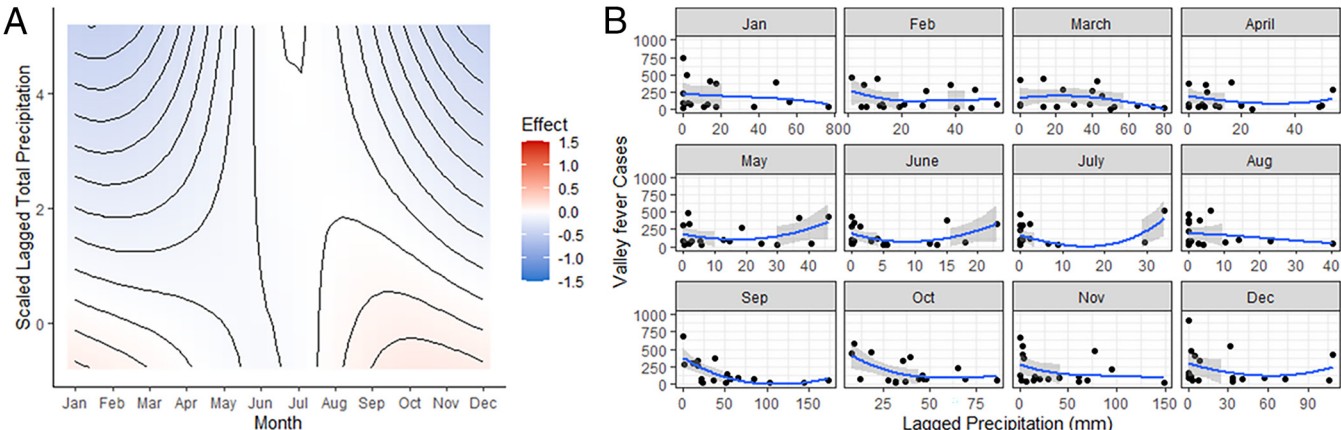

**FIG 3** The effects of lagged precipitation on the number of Valley fever cases. A 2-month lag time before the month where cases were reported was employed. (A) Simulation of predicted Valley fever cases across the entire range of lagged precipitation. Red means that there is a strong effect of maximum temperatures and Valley fever infections, and blue indicates a weak effect of maximum temperature on Valley fever infections. (B) Raw lagged precipitation data plotted against monthly Valley fever data added together from all three counties. LOESS smoother line used in order to visualize relationships.

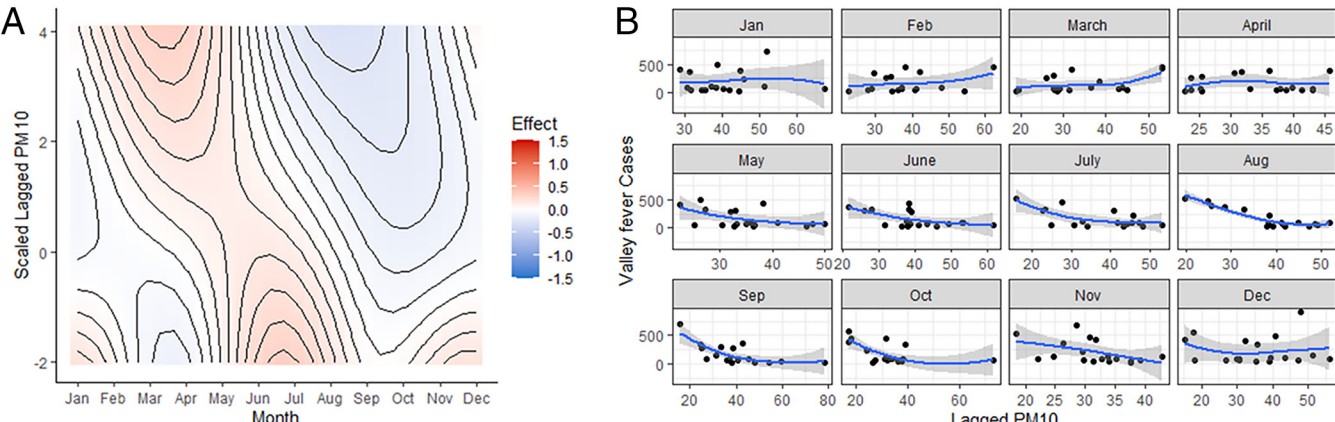

**FIG 4** The effects of lagged PM10 and temperature on the number of Valley fever cases. A 2-month lag time before the month where cases were reported was employed. (A) Simulation of predicted Valley fever cases across the entire range of lagged PM10. Red means that there is a strong effect of maximum temperatures and Valley fever infections, and blue indicates a weak effect of PM10 on Valley fever infections. (B) Raw lagged PM10 data plotted against monthly Valley fever data added together from all three counties. LOESS smoother line used in order to visualize relationships.

more important predictor. Lagged effects were also observed to be important by Kolivras and Comrie in 2003; however, the significant predictor variables were most sensitive in the winter months and lagged 1 year prior (20).

The effects of climate on the incidence of Valley fever in southern Arizona are quite complex with no obvious pattern when considering only the raw data. The effect of each climatic variable fluctuates throughout the year, and the effect is often nonlinear. The influence of lagged effects also appears to be important. Without the implementation of sophisticated statistical models, such as generalized additive models, quantifying these effects would be challenging. Binning case counts and climate variables by month disentangles the complexity of these relationships, and the effects are more pronounced. This may explain why historically it has been difficult to associate fungal infections with climate, and it emphasizes the complexity of the biology that underlies these climate associations.

**Influence of climate on the *Coccidioides* life cycle.** PM10 is a strong predictor of Valley fever cases because this particulate matter fraction contains fungal spores that cause infection. Particulate matter is an important gauge of air quality, and this metric quantifies the amount of pollutants and biological material such as allergens and potentially pathogenic fungi in the air (29). These small particles are associated with bacterial and fungal spores and are shown to have the most human health impacts (29, 30). These particles have the potential to be airborne for an extended period of time and can travel great distances (29). Several studies have shown that PM10 is an effective predictor of certain infectious diseases that are environmentally acquired (31–33). Our current study shows that PM10 in southern Arizona is a significant predictor of Valley fever cases but only during certain seasons. Previous studies have shown a relationship between soil-disturbing activities releasing a large quantity of airborne dust and Valley fever outbreaks (34–44), but this might not account for most cases that are caused by higher levels of fungi in the air column.

The climatic data were taken from NOAA land-based climate monitoring systems located in each of the representative counties in southern Arizona. There could be subtle differences between rural and urban locations that could skew the data and influence the results of the model. For future analysis, multiple locations from each county should be used to get an accurate representation of that location.

We hypothesize that the effect of wetting and drying on fungal growth and conidiation could explain why the 2-month lag in precipitation shows a strong effect on number of infections. The initial moisture leads to increased fungal biomass with the subsequent drying leading to increased airborne spores, which then have the potential for increased human infection. Precipitation is essential for growth and metabolism for all

soil fungi, but with potentially even greater effects on a desert fungus such as *Coccidioides* species. Water is essential for the transport of nutrients from the soil substrate into fungal cells as well as the hydrolytic process of breaking down macromolecules from the substrate to be used in metabolism (45). In semiarid and arid ecosystems, such as the Sonoran Desert, the soil experiences drought followed by rapid rewetting during the monsoon. The first 24 h after a wetting event has the greatest concentration of nutrient availability from the substrate as well as highest microbial activity followed by a drastic reduction in microbial activity and biomass as the soil dries (46). There is also a shift in the microbial soil community, which goes from being fungal dominant to Gram-positive bacterial dominant (46–48). The increase in moisture from infrequent desert rain may influence mycelium growth of *Coccidioides* spp., followed by a proliferation of conidial production with the prolonged dry periods leading to less nutrient availability. This increase in conidial production leads to an increase in infectious conidia in the environment and increased mammalian infections.

Understanding when a human population is at most risk of being exposed to *Coccidioides* has tremendous public health and economic implications. This model and others that come after it will allow public health departments to make more accurate public health announcements regarding risk for exposure. We have shown that we can use climatic variables (PM10, precipitation, and temperature) to accurately predict increases in Valley fever infections. By utilizing complex analytical methods developed for unrelated biological systems, such as generalized additive models, we demonstrate the utility in understanding the seasonal relationship of climate and Valley fever to predict risk.

## MATERIALS AND METHODS

**Arizona case data.** Monthly Valley fever case data were collected from the Arizona Department of Health Safety and the CDC (7). We focused on cases reported in Pima, Pinal, and Maricopa counties from 2013 to 2018. These counties are located in the region of hyperendemicity and have the highest reported cases of human disease in the state of Arizona (7). Reporting of Valley fever was mandatory in Arizona during this time frame, and an increase in cases was observed.

**Climatic data.** Climate data were obtained from the National Oceanic and Atmospheric Administration land-based stations from Maricopa, Pima, and Pinal counties in Arizona (49). Data were collected from one monitor system per county. Climate variables used for the Valley fever case model were taken from the month the cases were reported. The variables used in our analysis are wind speed (miles per hour [mph]), mean maximum temperature (°C), and total monthly precipitation (millimeters). Lagging variables simultaneously account for delays in diagnosis of disease (i.e., delay between exposure and case detection), as well as time for the fungal development. We implemented a 2-month lag period for covariates to account for delays in diagnosis and reporting. We treated the concurrent variables as different variables than lagged variables and tested for correlation via Pearson correlation.

For each covariate, we centered and scaled the climate variables by subtracting the mean of each data set from each individual data point and dividing all data points by the standard deviation of the data set.

Air quality particulate matter 10 $\mu$m and smaller (PM10) is commonly used to quantify biological particles in ambient air that are generally deposited in the upper respiratory tract and linked to certain infectious diseases (31, 50, 51). We are using PM10 as a proxy for airborne *Coccidioides* spores in the ambient atmosphere (15). This is measured in micrograms per cubic meter.

**Statistical analysis.** The effects of climate variables on case counts may change over time, and these effects may be nonlinear. For instance, thermal performance curves for microbial growth are often nonlinear (52). Therefore, we used generalized additive modeling (GAM). Specifically, GAMs use smoothed functional relationships (splines) to explain the potential nonlinear effects of covariates, and these splines can be two dimensional, allowing us to account for time-varying effects (26). Moreover, because the Valley fever case data are integer values, we assumed a negative-binomial error distribution for the model.

**Model structure.** We used the mgcv R package (53, 54) to perform our analyses (see Supplemental data file 1 in the supplemental material [R code] for details), and the full model used the following structure:

**(i) Full model including lagged variables.** Human Cases~te(Month, Precipitation)+te(Month, Temperature)+te(Month, Wind Speed)+te(Month, PM10)+te(Month, Lagged Precipitation)+te(Month, Lagged PM10)+s(County)+s(Year).

The above model structure represents the influence of each climatic covariate on the response variable (human cases of Valley fever). Each continuous covariate employs a cyclic cubic regression spline [i.e., the function, te()] to estimate the marginal effects of the covariate and month of the year. We chose this spline because of the cyclical nature of seasonally recurrent disease transmission. To account for nonindependence of the case data within years and counties, the effects of county and year were added as random effect splines [i.e., the function s()]. We reduced overfitting and limited the degrees of

freedom by running an iterative model to get the optimal smoothing parameter ($k = 5$). The gam.check function in the R package mgcv allows the user to determine how the selected $K$ (smoothing parameter) fits the predictors (overfit check).

**Model selection.** Each model was compared to a basic model in which no splines were used. This was a type of null model or "strawman," which we refer to as the "No Seasonality" model. In other words, no cyclic cubic regression splines were used in this simple "no seasonality" model. More complex models (models B, C, and D) included cyclic cubic regression splines and random effect splines along with the climate covariates (Table 1). Model E has lagged precipitation and lagged PM10 as covariates. We assessed model performance using multiple metrics: Akaike information criterion (AIC), explained deviance, adjusted $R$-squared, and root mean squared error, as well as examined for variable autocorrelation by Pearson correlation test. A summary of all covariates and splines of each test model can be found in Table S1 in the supplemental material.

**(i) Null model.** Human cases~Precipitation+Temperature+Wind Speed+PM10+Month+County.

## SUPPLEMENTAL MATERIAL

Supplemental material is available online only.

**SUPPLEMENTAL FILE 1**, PDF file, 1.8 MB.

## ACKNOWLEDGMENT

This work was supported by an Arizona Biomedical Research Center grant (ABRC 16-162415) to B.M.B.

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
