## [Reviewer comments · Microbiology Spectrum]

Microbiology Spectrum

PM10 and Other Climatic Variables are Important Predictors of Seasonal Variability of Coccidioidomycosis in Arizona

Daniel Kollath, Joseph Mihaljevic, and Bridget Barker

Corresponding Author(s): Bridget Barker, Northern Arizona University

Review Timeline:

Submission Date:	November 24, 2021
Editorial Decision:	January 4, 2022
Revision Received:	February 24, 2022
Accepted:	February 28, 2022

Editor: Christina Cuomo

Reviewer(s): The reviewers have opted to remain anonymous.

Transaction Report:

DOI: <https://doi.org/10.1128/spectrum.01483-21>

January 4, 2022

Dr. Bridget M Barker
Northern Arizona University
The Pathogen and Microbiome Institute
Applied Research & Development Building
Room 210 Building 56
Flagstaff, Arizona 86011-4073

Re: Spectrum01483-21 (**PM10 and Other Climactic Variables are Important Predictors of The Seasonality of Coccidioidomycosis in Arizona**)

Dear Dr. Bridget M Barker:

Thank you for submitting your manuscript to Microbiology Spectrum. Two reviewers have provided feedback that I would like you to address in a revision; one had reviewed the original submission, and the other is a new review. As this new reviewer notes in the attached file, please ensure that the methods are described in more detail and that the concern about overfitting of the model is addressed.

Link Not Available

Sincerely,

Christina Cuomo

Journals Department
Reviewer comments:

Reviewer #1 (Comments for the Author):

NONE

Reviewer #2 (Comments for the Author):

Please see the attached document for my review.

Staff Comments:

Preparing Revision Guidelines

Please return the manuscript within 60 days; if you cannot complete the modification within this time period, please contact me. If you do not wish to modify the manuscript and prefer to submit it to another journal, please notify me of your decision immediately so that the manuscript may be formally withdrawn from consideration by Microbiology Spectrum.

Review of “PM10 and Other Climatic Variables are Important Predictors of the Seasonality of Coccidioidomycosis in Arizona” by Kollath, Mihaljevic, and Barker

Valley fever is an understudied fungal disease and the majority of cases are reported in Arizona. Since the 1940s, it's been hypothesized that wet conditions allow *Coccidioides* to grow and dry conditions cause spores to be aerosolized—that's when animals can breathe them in and become infected with Valley fever. Because of this, short-term climate conditions (months/seasons) likely play a role in the variability of disease cases.

This analysis explores temperature, precipitation, and PM10 in relation to Valley fever cases from 2013-2018 to explore contemporary relationships between climate drivers and cases in a highly endemic region. This type of analysis has been previously reported in the literature (e.g., Comrie 2005), but it is beneficial to test to see if these relationships hold with contemporary data.

I believe this study will be a benefit to the Valley fever community and it's valuable to test whether old hypotheses still hold true. I have some concerns before the publication of this article:

As currently written, there needs to be more information in the methods section regarding the model development for clarity. There also needs to be a more robust discussion section to tie the results of this study to previous literature and move the science forward. The finding that PM is highly significant in predicting Valley fever cases is rather alarming, since no other study found this strong of a relationship. I'm suspect that there could be overfitting happening with the model, especially since there are a lot of outliers in the data plots shown. Please explain how you checked for overfitting. My comments and suggestions are organizing into major, minor, and figure.

Major comments and suggestions:

1. Introduction: The first paragraph is a bit broad given the topic of the paper is cocci. I suggest shortening to keep your reader focused. By starting on line 61, I get all the information I need about cocci.
2. Introduction line 88: I believe Comrie did create a forecast model? Maybe the motivation of your study is not that you're filling in the void of a forecast model, but that you are re-testing these hypothesis on more contemporary data?
3. Variable/model selection:
 - a. I don't understand the reasoning why the lagged variables and time-concurrent variables would be in the same model, if the hypothesis is that lagged variables would be more important. I suspect the lagged variables and time-concurrent variables are highly correlated and one would essentially “drop out” of the model anyways.
 - b. Why didn't you look at lagged temperatures since the combined effects of lower precipitation and higher temperatures in the same month would dry out the soils?
 - c. How did you limit the degrees of freedom in the splines to reduce overfitting? How did you test for overfitting?

- d. From how it's written, I don't understand where I can find the information about which variables went into models A-E.
4. Methods PM measurements: Where were the PM monitors located within the counties? Did you only use 1 station per county? Sometimes there are in the more urban areas, but other times there are out in very rural areas and may not be well representative of the population. Please include this information somewhere and mentioning this point in discussion may be worthwhile.
5. Time frame and language throughout: At some points the focus is on "seasons", but rather your models are predicting on the monthly time scale? Please clarify the time frame of your models and clarify throughout the text whether you mean season or month
6. You didn't state the conclusions of your two hypotheses proposed in your Introduction
7. Discussion: there have been a lot of mixed results in previous studies looking at PM10 versus Valley fever cases. There is no discussion of how your results relate to these previous studies and why yours might be so robust.

Minor comments and suggestions:

8. Throughout manuscript: change "climactic" to "climatic"
9. Title: Your analysis doesn't predict the seasonality (the normal annual cycle), but rather the monthly or annual variability, right? Suggest small phrasing change to "PM10 and Other Climatic Variables are Important Predictors of Coccidioidomycosis Cases in Arizona"
10. Throughout manuscript: I believe the community is shifting to use "Valley fever" with "fever" lowercase since it's named after the San Joaquin Valley of California (where "Valley" is part of a proper noun). I suggest changing to lowercase "fever" to follow disease name formatting (like West Nile fever, dengue fever).
11. Abstract line 21: Do the previous studies only look at annual patterns? I believe there are several that look at monthly relationships, too
12. Abstract line 32: "This study improves...": can you be more specific here?
13. Importance line 40: suggest replacing "but reported" with "and reported"
14. Introduction line 50: I don't think you need the ;
15. Introduction line 81: The focus in the last sentence was on meteorological patterns (or short term climate) affecting case counts and Weaver focuses on long-term trends, so I wouldn't include this study here for relevancy. It's focusing on two different questions.
16. Introduction line 88: Is there more reference information for the Lee citation, 19?
17. Introduction: suggest breaking up and restructuring the last paragraph into two for clarity and readability
18. Introduction: could you specify why there are multiple models? Did you break it up by seasons?
19. Introduction last paragraph: could you add in some citations for why temperature, precip, wind, etc. are important?
20. Introduction: you are examining lagged responses between climate variables and cases but didn't explain why (incubation time and reporting lags, right?). This would be helpful for the reader
21. Methods line 117-120: do you have references for these findings or is this based on your data you collected?

22. Methods line 128: suggest changing “In all cases” to “For each climate variable” since cases could mean Valley fever
23. Methods line 129: suggest changing “mean of all data” to “mean of each dataset”
24. Methods line 130: should “of all point” be “of the dataset”?
25. Methods line 163: How long is the lag?
26. Results line 169: was there evidence for bimodal seasonality like other papers suggest?
27. Discussion line 203: Incidence is usually defined as cases per population. You didn’t normalize the cases, right? So maybe rephrase this to just say “cases”?
28. Discussion line 235: seasons or months?
29. Figure 2: change VF to Valley fever

Figures

30. Figure 1: add PM10 units to plots
31. Figure 1A: I don’t quite understand how this figure was developed and what the contour lines represent. What does it mean to be simulated?
32. Figure 1B: are you adding all the counties data together here?
33. Figure S2: I thought the point of the model was to predict monthly cases, not the seasonal cycle? I don’t understand why the 2019 graph is helpful here unless you show what your model would have predicted for 2019.
34. Figure S1, S2: I don’t think these figures are cited in the text

Response to reviewer:

Reviewer 1

Major comments and suggestions:

1. Introduction: The first paragraph is a bit broad given the topic of the paper is cocci. I suggest shortening to keep your reader focused. By starting on line 61, I get all the information I need about cocci.

We started broad to appeal to a broad audience and because we wanted to address the severity of emerging dimorphic fungal pathogens. These pathogens are very understudied and approaches such as ours (and others) can be useful in learning about these neglected diseases.

2. Introduction line 88: I believe Comrie did create a forecast model? Maybe the motivation of your study is not that you're filling in the void of a forecast model, but that you are re testing these hypothesis on more contemporary data?

We agree and have changed some language to make it clear that we are testing to see if patterns previously described still hold true with the emergence of cases. These previous approaches to examine the impact of climate on Valley fever case variabilities have been informative; however, with the growing number of cases we must examine if these patterns still hold true as well as develop methods that can define climate variables that predict seasonal or yearly outbreaks of Valley fever. We have also added references for 2 new papers that were recently published (Chow 2021 and Comrie 2021)

3. Variable/model selection:

a. I don't understand the reasoning why the lagged variables and time-concurrent variables would be in the same model, if the hypothesis is that lagged variables would be more important. I suspect the lagged variables and time-concurrent variables are highly correlated and one would essentially "drop out" of the model anyways.

Thank you for asking this, we wanted to examine the lagged variables to get at the question of delayed diagnoses, meaning is an increase of cases in October attributed to an increase in precipitation in July. We treated the concurrent variables as different then the lagged variables and tested for correlation and is they were highly correlated they would be removed from the model but Lagged PM10 and precipitation were not highly correlated.

We implemented a two-month lag period to covariates to account for delays in diagnosis and reporting. We treated the concurrent variables as different variables then lagged variables and tested for correlation via Pearson correlation.

b. Why didn't you look at lagged temperatures since the combined effects of lower precipitation and higher temperatures in the same month would dry out the soils?

We did not include lagged temperature because we hypothesized that lagged precipitation is biologically more important. Statistically, we removed lagged temperature from the model structure due to strong autocorrelation with other variables.

c. How did you limit the degrees of freedom in the splines to reduce overfitting? How did you test for overfitting?

Added to methods. We reduced overfitting and limited the degrees of Freedom by running an iterative model to get the optimal smoothing parameter (k=5). The R package mgcv has a built in function called "gam.check" that allows you to check how the selected K (smoothing parameter) fits the predictors (over fit check)

d. From how it's written, I don't understand where I can find the information about which variables went into models A-E.

The "Model Selection" Section in the methods summarizes the process of picking the models. Each variable is also described in table 2. The model development process and information about each variable that went into each model is also summarized in table 1. We have added a supplemental table that we cited in the text that highlights each variable in each model. A summary of all covariates and splines of each test model can now be found in supplemental table 1.

4. Methods PM measurements: Where were the PM monitors located within the counties? Did you only use 1 station per county? Sometimes there are in the more urban areas, but other times there are out in very rural areas and may not be well representative of the population. Please include this information somewhere and mentioning this point in discussion may be worthwhile.

We used just one monitor/county. Great point, we have added to the discussion. The climatic data were taken from NOAA land-based climate monitoring systems that are located in each of the representative counties in southern Arizona. There could be differences between rural and urban locations that could skew the data and influence the results of the model. For future analysis multiple locations from each county should be used to get a larger representation of that location.

5. Time frame and language throughout: At some points the focus is on “seasons”, but rather your models are predicting on the monthly time scale? Please clarify the time frame of your models and clarify throughout the text whether you mean season or month

The timeframe of our models is on a monthly scale, we clarified this throughout the text. Binning case counts and climate variables by month disentangles the complexity of these relationships, and the effects are more pronounced

6. You didn’t state the conclusions of your two hypotheses proposed in your Introduction

It seems that temperature and precipitation do not have as much of an effect on cases as we hypothesized but rather PM10 is more important. Lagged effects were also observed to be important by Kolivras and Comrie 2003 however the significant predictor variables were most sensitive in the winter months and lagged one year prior [21]. Added in discussion.

7. Discussion: there have been a lot of mixed results in previous studies looking at PM10 versus Valley fever cases. There is no discussion of how your results relate to these previous studies and why yours might be so robust.

We addressed this and added the results from previous studies.

Response to reviewer:

Reviewer 2

Methods: information regarding model development and checks for model overfitting

Discussion: tie in previous results more robustly

Minor comments and suggestions:

8. Throughout manuscript: change “climactic” to “climatic”

Thank you for catching that, we replaced all.

9. Title: Your analysis doesn’t predict the seasonality (the normal annual cycle), but rather the monthly or annual variability, right? Suggest small phrasing change to “PM10 and Other Climatic Variables are Important Predictors of Coccidioidomycosis Cases in Arizona”

Great suggestion and you are correct about the seasonal variability

10. Throughout manuscript: I believe the community is shifting to use “Valley fever” with “fever” lowercase since it’s named after the San Joaquin Valley of California (where “Valley” is part of a proper noun). I suggest changing to lowercase “fever” to follow disease name formatting (like West Nile fever, dengue fever).

Replaced

11. Abstract line 21: Do the previous studies only look at annual patterns? I believe there are several that look at monthly relationships, too

Previous studies that look at annual and monthly relationships of coccidioidomycosis and climate suggest that infection numbers are linked with precipitation and temperature fluctuations; however, these analytic methods may miss important non-linear, non-monotonic seasonal relationships between the response (Valley fever cases) and explanatory variables (climate) influencing disease outbreaks.

12. Abstract line 32: "This study improves...": can you be more specific here?

This study builds on and retests relationships described by previous studies regarding climate variables that are important for predicting risk of infection and understanding the biology of this fungal pathogen. We have clarified.

13. Importance line 40: suggest replacing "but reported" with "and reported"

Replaced

14. Introduction line 50: I don't think you need the ;

Taken out

15. Introduction line 81: The focus in the last sentence was on meteorological patterns (or short term climate) affecting case counts and Weaver focuses on long-term trends, so I wouldn't include this study here for relevancy. It's focusing on two different questions.

Agreed we removed that citation in that location.

16. Introduction line 88: Is there more reference information for the Lee citation, 19?

Lee, Chan Mi. *Spatiotemporal association between valley fever and PM10: a case study of Arizona*. Diss. 2020.

17. Introduction: suggest breaking up and restructuring the last paragraph into two for clarity and readability

Agreed. We separated into different paragraphs.

Our overall goal is to understand the seasonal dynamics of human infections and how inter-annual variation in climate will affect the risk of increased cases by retesting what others have previously observed with more contemporary data and analysis methods. We chose our climatic covariates based on previous studies, as well as our own hypotheses. Temperature and precipitation are likely biologically important to the lifecycle to the fungus [17, 20-22].

It has been hypothesized that inclement weather such as dust storms increases the risk of infection by *Coccidioides* although recent analysis shows no such connection we expect, wind speed to be important for explaining human disease because of the dispersal of the infectious fungal propagules that establish infection [18, 23-25]. Moreover, we hypothesize that the concentration of inhalable particulate matter, quantified as PM10, is important because infectious spores could be dispersed via dust. Finally, we accounted for delays in diagnosis by introducing lagged effects (we assume that there is time between

when a patient gets infected, when symptoms occur, and that case gets reported, so lagged variables are implemented to account for that).

We hypothesize that 1) seasonal temperature changes will impact fungal biomass in the environment (via growth and conidiation) leading to fluctuations in human disease, and we therefore expect increased cases with increased temperature in the summer months, and 2) because precipitation will impact soil moisture and enhance fungal growth, greater total precipitation in the winter and during the monsoon season will lead to increased infections, but this effect could be delayed until the desert soil dries out and conidia can be more efficiently dispersed by wind.

18. Introduction: could you specify why there are multiple models? Did you break it up by seasons?

We built and tested multiple models to test which is the best performing. For example, we built a very simple model that did not incorporate any splines (no seasonality model) and then added covariates to see which model performs the best. We concluded that the model with all covariates (including the lagged variables) was the best performing model. That is what is meant by “models”. We added the following sentence to clarify. These covariates were put into models and evaluated for importance for model performance, the highest performing model was used for the final analysis.

19. Introduction last paragraph: could you add in some citations for why temperature, precip, wind, etc. are important?

We added additional citations for these.

Comrie AC. Climate factors influencing coccidioidomycosis seasonality and outbreaks. *Environmental health perspectives*. 2005;113(6):688-92.

Gorris M, Cat L, Zender C, Treseder K, Randerson J. Coccidioidomycosis dynamics in relation to climate in the southwestern United States. *GeoHealth*. 2018;2(1):6-24.

Kolivras KN, Comrie AC. Modeling valley fever (coccidioidomycosis) incidence on the basis of climate conditions. *Int J Biometeorol*. 2003;47(2):87-101. doi: 10.1007/s00484-002-0155-x. PubMed PMID: 12647095.

Kolivras KN, Johnson PS, Comrie AC, Yool SR. Environmental variability and coccidioidomycosis (valley fever). *Aerobiologia*. 2001;17(1):31-42.

20. Introduction: you are examining lagged responses between climate variables and cases but didn't explain why (incubation time and reporting lags, right?). This would be helpful for the reader

Thank you for that observation and yes you are correct. Our thought process was to implement lagged variables to account for the delay between infection and reporting. We added information that explains that better. We assume that there is time between when a patient gets infected and when symptoms occurs and that case gets reported so lagged variables are implemented to account for that

21. Methods line 117-120: do you have references for these findings or is this based on your data you collected?

These findings were based off the below citation and added

Benedict K, McCotter OZ, Brady S, Komatsu K, Sondermeyer Cooksey GL, Nguyen A, et al. Surveillance for Coccidioidomycosis - United States, 2011-2017. MMWR Surveill Summ. 2019;68(7):1-15. doi: 10.15585/mmwr.ss6807a1.

These counties are located in the hyper-endemic region, and have the highest reported cases of human disease in the state of Arizona [11]

22. Methods line 128: suggest changing “In all cases” to “For each climate variable” since cases could mean Valley fever

Thank you for catching that, which could be confusing. We clarified this within the sentence. For each covariate, we centered and scaled the climate variables by subtracting the mean of each dataset from each individual data point and dividing all data points by the standard deviation of the dataset.

23. Methods line 129: suggest changing “mean of all data” to “mean of each dataset”

Changed

24. Methods line 130: should “of all point” be “of the dataset”?

Yes, changed

25. Methods line 163: How long is the lag?

2 months. We made that more explicit in the text. We implemented this two-month lag period to covariates to account for delays in diagnosis and reporting

26. Results line 169: was there evidence for bimodal seasonality like other papers suggest?

We found evidence of bimodality in the time we sampled (the pattern is clearer in some counties more than others) see the supplemental figs. We made it more explicit and referenced the sup figs in that paragraph. There is a pattern of bimodal seasonality in the three sampled Arizona counties (S1 and S2). In other words, Valley fever cases fluctuate throughout the year based on climate. Specifically, cases tend to rise during the summer months and winter months and decrease during the spring but show a slight increase in the fall months.

27. Discussion line 203: Incidence is usually defined as cases per population. You didn’t normalize the cases, right? So maybe rephrase this to just say “cases”?

Correct we did not normalize, we fixed this terminology

This study found significant monthly effects of climate on cases of coccidioidomycosis

28. Discussion line 235: seasons or months?

By month, we made that more clear

29. Figure 2: change VF to Valley fever

Made this consistent in all the figure legends

Figures

30. Figure 1: add PM10 units to plots

We added the units in the figure legend to make clear

*31. Figure 1A: I don't quite understand how this figure was developed and what the contour lines represent. What does it mean to be simulated?

This graph is exploring the relationship between month and PM10 and how these combination effects cases. By simulation we meant that we were using the model to predict across a range of PM10, I can remove "simulation" so that is clearer. The contour lines are identifying combinations of PM10 values and month and the effect the combination has on cases. These contour plots are model after this paper <https://peerj.com/articles/6876/> (see figure 8)

Figure 1a explores the relationship between month and PM10 and how in combination effects case numbers. The model is predicting across a range of PM10. The contour lines are identifying combinations of PM10 values and month and the relationship on Valley fever cases. We have added some language to better help the reader.

32. Figure 1B: are you adding all the counties data together here?

Yes the three counties are added together to produce this plot. We made it explicit in each of the figure legends for figures. Raw monthly PM10 data plotted against monthly Valley fever data added together from all three counties.

33. Figure S2: I thought the point of the model was to predict monthly cases, not the seasonal cycle? I don't understand why the 2019 graph is helpful here unless you show what your model would have predicted for 2019.

This figure was simply to show the effectiveness of predicting cases into the future (i.e. 2019) and what it would have predicted for this particular year. The effectiveness of the model to predict into the future was assed using 2019 Valley fever case data (S2). We did not use 2019 data in the development of the model. Due to COVID, no additional years are yet available.

34. Figure S1, S2: I don't think these figures are cited in the text

We cite these figures in the results when we bring up the bimodal pattern of disease that observed See above comment 26

February 28, 2022

Dr. Bridget M Barker
Northern Arizona University
The Pathogen and Microbiome Institute
Applied Research & Development Building
Room 210 Building 56
Flagstaff, Arizona 86011-4073

Re: Spectrum01483-21R1 (PM10 and Other Climatic Variables are Important Predictors of Seasonal Variability of Coccidioidomycosis in Arizona)

Dear Dr. Bridget M Barker:

Your manuscript has been accepted, and I am forwarding it to the ASM Journals Department for publication. You will be notified when your proofs are ready to be viewed.

Sincerely,

Christina Cuomo
Editor, Microbiology Spectrum
